# Sleep spindles mediate hippocampal-neocortical coupling during long-duration ripples

**Hong-Viet Ngo[1,2], Juergen Fell[3], Bernhard Staresina[1]\***

[1]School of Psychology and Centre for Human Brain Health, University of Birmingham, Birmingham, United Kingdom; [2]Donders Institute for Brain, Cognition and Behaviour, Radboud University Medical Centre, Nijmegen, Netherlands; [3]Department of Epileptology, University of Bonn, Bonn, Germany

**Abstract** Sleep is pivotal for memory consolidation. According to two-stage accounts, memory traces are gradually translocated from hippocampus to neocortex during non-rapid-eye-movement (NREM) sleep. Mechanistically, this information transfer is thought to rely on interactions between thalamocortical spindles and hippocampal ripples. To test this hypothesis, we analyzed intracranial and scalp Electroencephalography sleep recordings from pre-surgical epilepsy patients. We first observed a concurrent spindle power increase in hippocampus (HIPP) and neocortex (NC) time-locked to individual hippocampal ripple events. Coherence analysis confirmed elevated levels of hippocampal-neocortical spindle coupling around ripples, with directionality analyses indicating an influence from NC to HIPP. Importantly, these hippocampal-neocortical dynamics were particularly pronounced during long-duration compared to short-duration ripples. Together, our findings reveal a potential mechanism underlying active consolidation, comprising a neocortical-hippocampal-neocortical reactivation loop initiated by the neocortex. This hippocampal-cortical dialogue is mediated by sleep spindles and is enhanced during long-duration hippocampal ripples.

**\*For correspondence:**
b.staresina@bham.ac.uk

**Competing interests:** The authors declare that no competing interests exist.

## Introduction

Information transfer during sleep is thought to rely on systematic interactions of the cardinal non-rapid-eye-movement (NREM) sleep rhythms (*Diekelmann and Born, 2010*; *Klinzing et al., 2019*): First, the cortical <1 Hz slow oscillation (SO) opens windows of neuronal excitability and inhibition (up- and down-states, respectively) in cortical and subcortical regions (*Isomura et al., 2006*; *Timofeev, 2011*). Triggered by SOs, the thalamus generates sleep spindles, that is, transient (0.5–2 s) oscillatory activity between 12 and 16 Hz, via thalamo-cortical loops (*Timofeev and Steriade, 1996*). Nested in SO up-states, spindles gate $Ca^{2+}$ influx into dendrites and promote synaptic plasticity (*Astori et al., 2013*; *Fernandez and Lüthi, 2020*; *Niethard et al., 2018*; *Seibt et al., 2017*). Importantly, spindles have also been shown to group hippocampal sharp-wave ripples (SW-Rs) (*Siapas and Wilson, 1998*; *Sirota et al., 2003*; *Staresina et al., 2015*). SW-Rs are transient network oscillations in the hippocampal CA1 subfield emerging from recurrent interactions within CA3 and consist of a brief 80–200 Hz ripple-burst superimposed on a sharp wave, with the latter characterized by a peak frequency of ~3 Hz in humans (*Axmacher et al., 2008*; *Helfrich et al., 2019*; *Jiang et al., 2019a*; *Norman et al., 2019*; *Staresina et al., 2015*). Ripples have been linked to the reactivation of cell assemblies engaged during previous encoding (*Diba and Buzsáki, 2007*; *Dupret et al., 2010*; *Wilson and McNaughton, 1994*), and experimental suppression of ripples leads to an impairment of memory performance (*Ego-Stengel and Wilson, 2010*; *Girardeau et al., 2009*). Given that spindles group SW-Rs and induce neural plasticity (*Jiang et al., 2019a*; *Niethard et al., 2018*; *Seibt et al., 2017*; *Staresina et al., 2015*), they appear ideally suited to facilitate memory

consolidation mechanistically. That is, spindles might synchronize the hippocampus (sender) and cortical target sites (receiver) during reactivation events and thereby induce synaptic changes in neocortex for long-term storage (*Rosanova and Ulrich, 2005*; *Sejnowski and Destexhe, 2000*). However, whether and how spindles mediate the hippocampal-neocortical dialogue still remains poorly understood.

## Results

We here examined whole-night sleep Electroencephalography (EEG) recordings from neocortex and hippocampus in 14 pre-surgical epilepsy patients (*Figure 1*). In a first step, we algorithmically detected spindles in neocortex (NC; scalp electrode Cz) and posterior hippocampus (HIPP; *Staresina et al., 2015*) as well as hippocampal sharp-wave ripples (SW-Rs). Across participants, a total of 17,174 NC spindles, 14,748 HIPP spindles and 3,748 HIPP SW-Rs were identified (*Figure 1*). For sleep characteristics and patient-specific event numbers and densities (events per minute), see *Tables 1* and *2*, respectively.

### Cortical and hippocampal spindles around SW-Rs

To examine whether SW-Rs co-occur not only with spindles in HIPP (*Jiang et al., 2019a*; *Staresina et al., 2015*) but also in NC, we first derived time-frequency representations (TFRs) time-locked to discrete HIPP SW-Rs (*Figure 2*). Results were statistically compared to TFRs obtained from control events, that is, matched ripple-free intervals randomly drawn from NREM sleep. In HIPP, we found an extended cluster of significant power increases encompassing two distinct components (p=0.001, *Figure 2A*): First, an increase in spindle power (12–18 Hz), peaking at ~300 ms after SW-Rs. Second, a more widespread frequency cluster with a maximum at 3 Hz, likely reflecting the sharp-wave component (*Axmacher et al., 2008*; *Oliva et al., 2018*; *Staresina et al., 2015*). Critically, when locking the TFR in NC to SW-Rs in HIPP (*Figure 2B*), we again observed significant power increases compared to ripple-free control events. A significant cluster emerged from 11 to 16 Hz (p=0.001), again peaking shortly after SW-Rs. The concurrent spindle power increase in both regions (*Figure 2C*) around HIPP SW-R might indicate a role of spindles in mediating cortical-hippocampal communication (see next section).

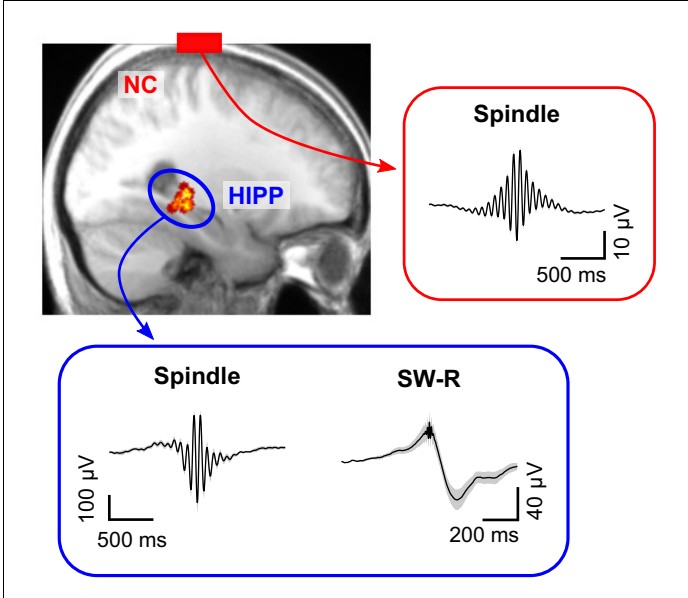

**Figure 1.** Cortical sleep spindles and hippocampal spindles and SW-Rs. *Top left*: heat map illustrating the position of individual contacts across all patients within the hippocampus overlaid on a sagittal slice of the mean structural MRI. *Right* and *bottom* insets show the grand average (± SEM) spindle and sharp-wave ripple waveforms across all patients in neocortex (red, NC) and hippocampus (blue, HIPP).

**Table 1.** Sleep architecture.
Mean ± SEM proportion of sleep stages S1, S2, slow wave sleep (SWS) and rapid eye movement (REM) sleep relative to the total time spent asleep.

|  | Mean | SEM |
|---|---|---|
| S1 (%) | 22.8 | 4.6 |
| S2 (%) | 44.0 | 3.3 |
| SWS (%) | 16.4 | 2.6 |
| REM (%) | 16.8 | 2.2 |
| Total sleep time (min) | 427.6 | 29.0 |

Regarding event contingencies,~5% and 7% of all algorithmically detected sleep spindles in NC and HIPP, respectively, occurred within ±1 s of HIPP ripple maxima. Conversely, ~17% and 22% of HIPP ripples exhibited an overlap with NC and HIPP sleep spindles, respectively (see also *Staresina et al., 2015*). This highlights that only a fraction spindles coincides with HIPP ripples, raising the interesting question whether the remainder of spindles might serve different functions. That said, the precise numbers of event overlap depend on the detection thresholds for spindles and ripples and, for the latter, might have been reduced by the exclusion of physiological ripples coinciding with artifactual, epileptic activity. Moreover, it is likely that a certain amount of NC spindles overlap with ripples emerging in parts of HIPP not captured by our chosen contact.

While theoretical and empirical work also implicates SOs/slow-waves in the hippocampal-neocortical dialogue (*Maingret et al., 2016*; *Sejnowski and Destexhe, 2000*; *Sirota et al., 2003*), our ripple-locked TFR analyses revealed no significant effect at ~1 Hz. However, this might result from our

**Table 2.** Properties of sleep spindles and sharp-wave ripples.
Patient-specific count and density (events per minute) of algorithmically detected spindles and sharp-wave ripples (SW-Rs) as well as the corresponding mean (± SEM) across participants. Note that event densities reflect number of events relative to artifact-free NREM time, resulting in possibly different densities despite similar event counts.

| Patient | Spindle count | | Spindle density (per min) | | SW-R count | SW-R density (per min) |
|---|---|---|---|---|---|---|
|  | NC | HIPP | NC | HIPP |  |  |
| 1 | 1440 | 1296 | 6.3 | 6.4 | 135 | 0.7 |
| 2 | 1243 | 571 | 5.5 | 4.3 | 107 | 0.8 |
| 3 | 1703 | 1725 | 5.5 | 5.9 | 116 | 0.4 |
| 4 | 1074 | 978 | 3.0 | 3.1 | 626 | 2.0 |
| 5 | 1427 | 625 | 5.4 | 2.4 | 206 | 0.8 |
| 6 | 1001 | 691 | 4.0 | 3.9 | 153 | 0.9 |
| 7 | 903 | 1975 | 3.3 | 6.7 | 403 | 1.5 |
| 8 | 1004 | 892 | 4.9 | 4.5 | 400 | 2.1 |
| 9 | 2079 | 2038 | 6.2 | 6.0 | 518 | 1.6 |
| 10 | 1120 | 840 | 6.3 | 4.3 | 245 | 1.4 |
| 11 | 494 | 739 | 4.7 | 7.0 | 208 | 2.0 |
| 12 | 448 | 417 | 5.6 | 6.3 | 44 | 0.7 |
| 13 | 1339 | 406 | 5.6 | 2.0 | 45 | 0.2 |
| 14 | 1899 | 1555 | 6.5 | 6.1 | 542 | 2.1 |
| Mean | **1226.7** | **1053.4** | **5.2** | **4.9** | **267.7** | **1.2** |
| SEM | **125.9** | **150.6** | **0.3** | **0.4** | **50.0** | **0.2** |

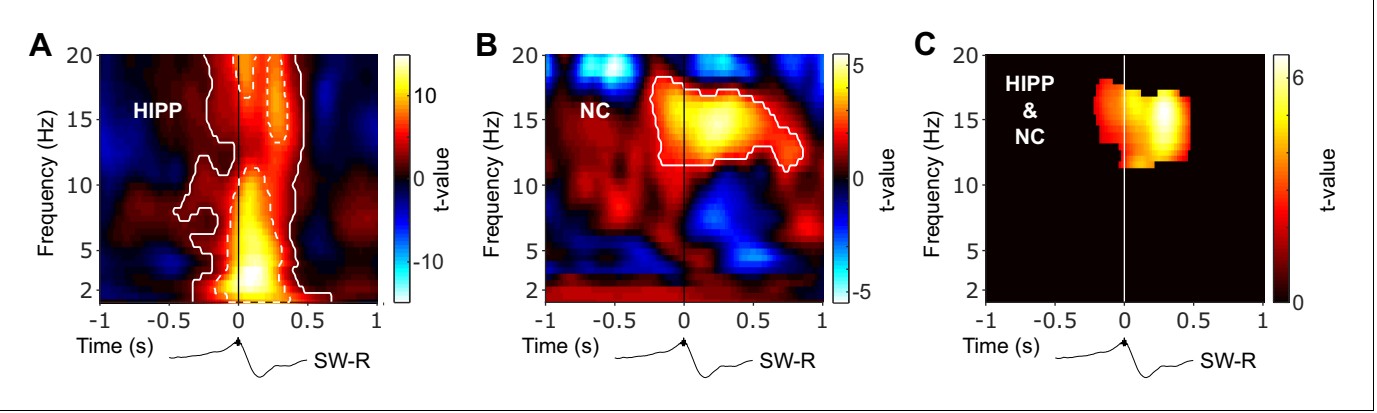

**Figure 2.** Hippocampal and neocortical spindle activity coincides during SW-Rs. Statistical maps (t-values) contrasting ripple-locked vs. control TFRs within HIPP (**A**) and NC (**B**). Hot colors indicate power increases around SW-Rs, whereas cold colors indicate relative power decreases. White contours mark significant clusters obtained from a cluster-based permutation procedure. (**C**) Significance mask derived from the overlap of significant clusters between NC and HIPP. Color represents the mean t-value from the corresponding statistical masks. Black traces below TFRs illustrate the timing of power changes relative to SW-Rs. Of note, dashed white contours in (**A**) represent significant clusters based on a highly conservative statistical threshold (cluster alpha-level $\alpha = 10^{-11}$) to further illustrate the distinct 3 Hz sharp-wave and 12–20 Hz spindle clusters.

The online version of this article includes the following figure supplement(s) for figure 2:

**Figure supplement 1.** Power in the Slow Oscillation (SO) band.

control procedure, which extracts matched NREM epochs without HIPP SW-Rs. That is, the prevalence of SOs throughout NREM sleep (and thus also in control epochs) might obscure SO-related effects aligned to SW-Rs. Corroborating this notion, comparing ripple-locked and NREM control event TFRs with control events from REM sleep confirmed a significant global increase in SO power (1–1.5 Hz) during NREM both in NC and HIPP (ripple-locked vs. REM control events: $t_{(3617)} > 11.441$, p<0.001; NREM control vs. REM control events: $t_{(3974)} > 54.177$, p<0.001, *Figure 2—figure supplement 1*). No difference was seen between ripple-locked events and NREM control events ($t_{(3747)} < 1.559$, p>0.118). These results suggest that any clustering of SOs around hippocampal ripples might be concealed by the strong prevalence of SOs throughout NREM sleep. That said, these results might also indicate that SOs do not mediate hippocampal-cortical interactions around ripples directly, but indirectly by coordinating sleep spindles (*Mölle et al., 2011*; *Staresina et al., 2015*). Indeed, a recent human iEEG study showed that coupling between prefrontal cortex spindles and medial temporal lobe ripples was greater when prefrontal spindles were nested in the up-state of concomitant SOs (*Helfrich et al., 2019*). Similarly, spindle coupling between neocortex and hippocampus was found to be increased at particular SO-phases (*Cox et al., 2020*). This coordinating role of SOs notwithstanding, our results emphasize that spindles group around hippocampal ripples above and beyond background NREM spindle activity.

## Cortical-hippocampal spindle coupling around SW-Rs

Next, we asked whether the cross-regional co-activation in the spindle band around HIPP SW-Rs may indeed reflect an increase in functional coupling between NC and HIPP. To this end, we calculated ripple-locked spectral coherence between NC and HIPP. Based on the concurrent power increases shown in *Figure 2C*, we determined coherence specifically for the observed spindle cluster around SW-Rs (for extended time-frequency resolved coherence analysis, see *Figure 3—figure supplement 1*). As shown in *Figure 3A*, spindle-band coherence between NC and HIPP was indeed significantly increased compared to control events (z-value = 2.12, p=0.034). Examination of time-resolved 12–16 Hz coherence confirmed a significant increase reaching its maximum shortly after the SW-R (*Figure 3B*). Identical results were obtained using amplitude- or phase-based connectivity measures (*Figure 3—figure supplement 2A–D*), that is, orthogonalized power correlation (*Hipp et al., 2012*) and phase-locking value (*Lachaux et al., 1999*), which validated the robustness of our coherence-based approach (which is impacted by both amplitude and phase) and - more importantly - ruled out any influence of volume conduction or spurious correlations due to a

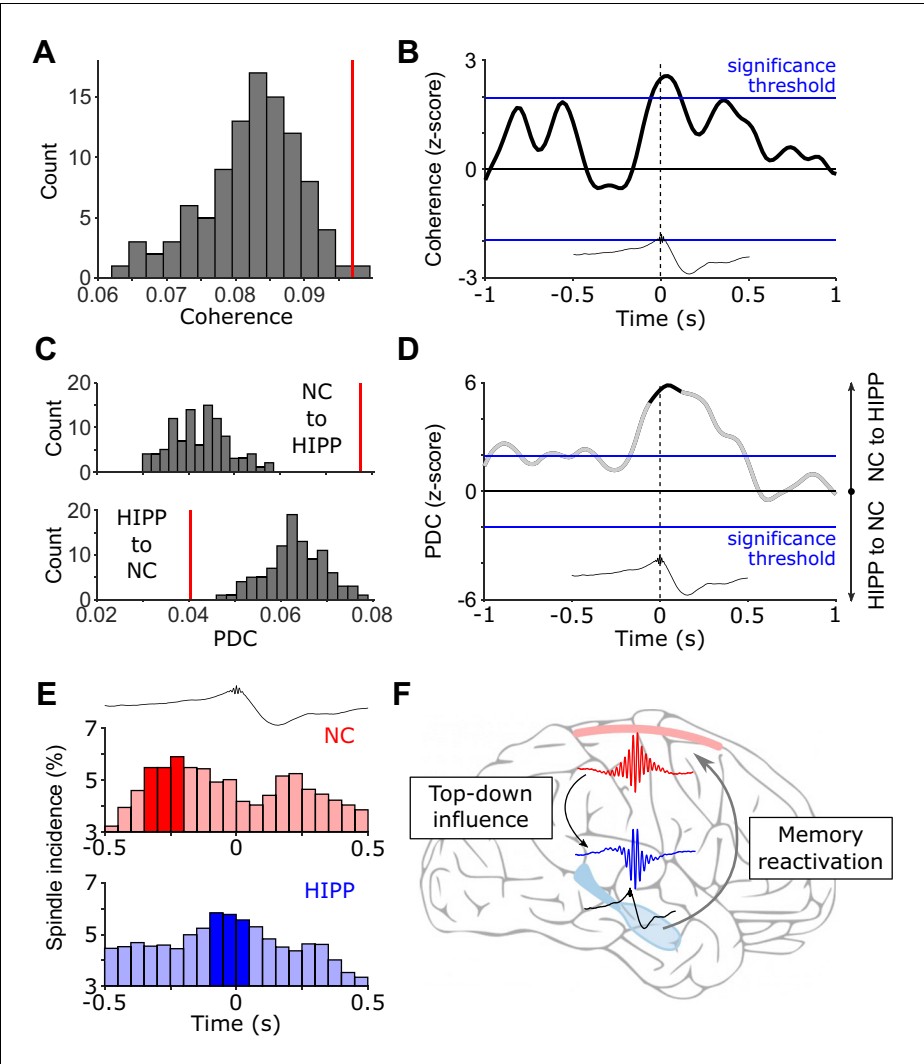

**Figure 3.** (Directional) Cortical-hippocampal communication via spindles. (**A**) HIPP-NC coherence. The red line depicts the observed coherence between HIPP and NC in the cluster of concurrent increases in spindle power around HIPP SW-Rs (see *Figure 2C*). Histogram depicts the distribution of individual coherence values for 100 sets of control events. (**B**) Time-resolved HIPP-NC coherence for the 12–16 Hz spindle range transformed into a z-score with respect to the control events. Blue lines indicate standard significance thresholds (z = 1.96). Time 0 denotes HIPP SW-Rs. (**C**) Partial directed coherence (PDC). Red vertical lines represent directional influence from NC to HIPP (*top*) or directional influence from HIPP to NC (*bottom*) in the cluster of concurrent increases in spindle power around HIPP SW-Rs (*Figure 2C*). Histograms depict distribution of individual PDC values for 100 sets of control events. (**D**) Time course of difference between z-transformed NC - > HIPP and HIPP - > NC influence in the 12–16 Hz spindle range. Positive values signify a cortical influence in the 12–16 Hz spindle range on HIPP and vice versa for negative values. Black-colored sections correspond to time intervals with significant spectral coherence shown in (**B**). Blue lines mark standard significance thresholds (z = 1.96). (**E**) Peri-event histograms of spindle onsets in NC (*top*) and HIPP (*bottom*) within a ± 0.5 s time window around SW-Rs (time = 0 s, top trace). Dark colored bars indicate significant time bins, resulting from comparison with ripple-free control events (z > 1.96). (**F**) Schematic illustrating the hypothesized spindle-mediated cortical-hippocampal dialogue around SW-Rs: First, sleep spindles mediate a top-down influence from NC to HIPP. In HIPP, sleep spindles in turn coordinate the occurrence of SW-Rs on a fine temporal scale. SW-Rs are linked to the reactivation of relevant memory traces, thought to be distributed to neocortical sites for long-term storage (*Klinzing et al., 2019*).

The online version of this article includes the following figure supplement(s) for figure 3:

**Figure supplement 1.** Mutual cortical-hippocampal connectivity around SW-Rs.
**Figure supplement 2.** Amplitude- and phase-based cortical-hippocampal connectivity.
**Figure supplement 3.** Directed cortical-hippocampal connectivity around SW-Rs.

common referencing scheme. Note though that while power correlations and phase-locking are differentially sensitive to amplitude- and phase covariations, respectively (*Siegel et al., 2012*), they are not entirely independent (*Palva and Palva, 2018*; *Siems and Siegel, 2020*). In any case, these findings corroborate the notion that sleep spindles co-occurring in NC and HIPP around hippocampal ripples reflect an increase in cortical-hippocampal communication.

## Directional influence of cortical to hippocampal spindles prior to SW-Rs

After revealing enhanced spindle-mediated coupling between hippocampus and neocortex around HIPP ripples, we asked whether this interaction is directional, that is, do cortical spindles influence hippocampal spindles or vice versa? As a measure of directionality, we used partial directed coherence (PDC). Extending the concept of coherence, that is, an assessment of mutual synchrony based on phase and amplitude, PDC examines the relative timing of these features between two regions of interest. This approach allows quantifying whether the current state of a target region may be influenced by the past of the other region by taking into account the predictive information of both, the past of the other region and the past of the target region itself (*Baccalá and Sameshima, 2001*). Calculating PDC again for the cluster of concurrent HIPP-NC spindle power increase around SW-Rs (see *Figure 2C*; for extended time-frequency resolved PDC analysis see *Figure 3—figure supplement 3*) revealed an increase in directional influence primarily from neocortex to the hippocampus in comparison to control events (*Figure 3C*, z = 4.76, p<0.001). The inverse directionality (HIPP -> NC) was diminished around SW-Rs in comparison to control events (z = −3.17, p=0.002). The direct comparison of the directional influence between cortical and hippocampal spindles revealed an almost two-fold influence of neocortical spindles on hippocampal spindles than vice versa. Inspecting the temporal dynamics of PDC within the spindle range suggests that the top-down influence sets in before the occurrence of HIPP SW-Rs (*Figure 3D*). To further pinpoint the origin of the directional influence of NC on HIPP spindles, we extracted the onset latencies of discrete spindles in both regions with respect to HIPP SW-Rs. Note that while the TFR analysis shown in *Figure 2* highlighted significant increases in amplitude, this analysis is particularly geared toward discrete spindle events obtained via algorithmic detection methods. One advantage of this approach is that it bypasses some of the temporal ambiguities of TFRs due to filter smearing (*Iemi et al., 2017*). The histograms of spindle onset times with respect to hippocampal ripples (*Figure 3E*) revealed a maximum (significantly increased in comparison to control events) in both regions before the SW-Rs. Importantly though, this maximum occurred from −250 to −200 ms prior to the hippocampal ripple in NC and from −100 to −50 ms prior to the ripple in HIPP. Together, these results suggest a directional influence of NC spindles on HIPP spindles prior to the SW-R. It deserves explicit mention though that conclusive evidence for a driving or causal role of NC spindles would require experimental perturbation of the system. Therefore, interpretive caution is warranted for results from directional connectivity analyses as employed here.

## Hippocampal-neocortical coupling emerges preferentially during long-duration ripples

Our study is guided by the tacit assumption that ripple-locked hippocampal-neocortical interactions are instrumental for memory consolidation. However, without an explicit pre-sleep learning and post-sleep testing component, this assumption is rather conjectural. Interestingly though, while the critical role of ripples for memory-related processes has been established in rodents (*Ego-Stengel and Wilson, 2010*; *Girardeau et al., 2009*), a recent study further demonstrated that longer-duration ripples bear greater relevance for memory-related processes than short-duration ripples (*Fernández-Ruiz et al., 2019*). That is, replay of newly encoded information was associated with longer duration ripples and experimental ripple prolongation or perturbation of the late ripple component enhanced or diminished memory performance, respectively. We therefore reasoned that if our results reflect hippocampal-neocortical interactions in service of memory consolidation, they should be more pronounced for longer duration compared to shorter duration hippocampal ripples.

To assess this possibility, we divided, separately for each patient to ensure matching event numbers, all ripple events into tertiles based on their duration and categorized the first tertile as short-duration and the third tertile as long-duration SW-Rs (mean ± SEM tertile thresholds = 0.046 ± 0.001 s and 0.059 ± 0.002 s for short and long-duration SW-Rs, respectively, *Figure 4A*). Examining the

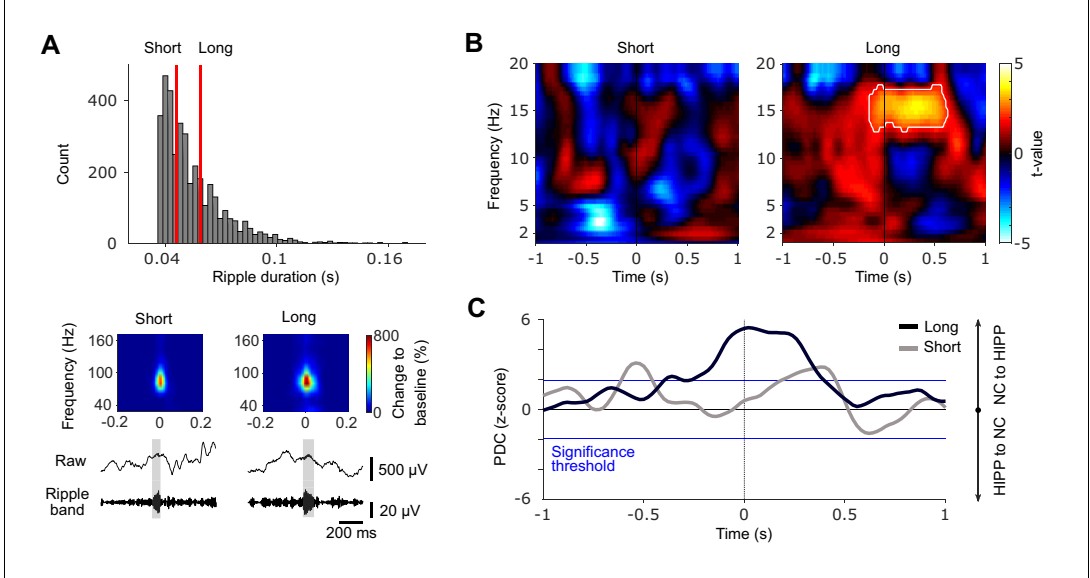

**Figure 4.** Cortical-hippocampal communication is enhanced during long ripples. (**A**) *Top*: histogram of ripple durations pooled across all patients. Red vertical lines indicate the average cut-offs for short (mean ± SEM duration=0.046 ± 0.001 s) and long ripple events (mean ± SEM duration=0.059 ± 0.002 s), based on patient-specific tertiles (short ripples: first tertile; long ripples: third tertile). *Bottom*: Grand average rippled-locked time-frequency representations (TFRs) as well as example traces (raw and filtered between 80 and 120 Hz) exemplifying short and long ripples. Color bar represents percentage change to a pre-event baseline from −2 to −1.5 s. (**B**) Statistical maps (t-values vs. control events) for neocortical TFRs (NC) locked to short (left) or long (right) hippocampal ripples. Hot colors indicate power increases around SW-Rs, whereas cold colors indicate relative power decreases. White contours mark significant clusters obtained from a cluster-based permutation procedure (p<0.05, corrected). (**C**) Time course of partial directed coherence (PDC; z-transformed vs. control events) in the 12–16 Hz spindle range locked to short (gray) and long (black) ripples. Positive values signify a cortical to hippocampal information flow and vice versa for negative values. Blue lines mark standard significance thresholds (z = 1.96; p<0.05).

neocortical TFR locked to different-duration hippocampal ripples again revealed a significant increase in spindle power with respect to matched control events, but only for long-duration ripples (*Figure 4B*). In fact, directly contrasting power values in the joint NC/HIPP spindle cluster (*Figure 2C*) around long- vs. short-duration ripples revealed greater NC spindle power during longer ripples ($t_{(2937)}$ = 1.76, p=0.039, one-sided independent samples t-test in accordance with *Fernández-Ruiz et al., 2019*). The same held true for the comparison of hippocampal spindle power locked to long- vs. short-duration ripples ($t_{(2937)}$ = 4.68, p<0.001, one-tailed). Additionally, we correlated – for each patient separately – ripple duration with spindle power in the joint spindle cluster (*Figure 2C*) across events in both NC and HIPP. Resulting correlation values were Fisher z-transformed and tested against zero via a one-sample t-test across patients. Correlation values were reliably positive in both regions (NC: average Spearman r = 0.05, $t_{(13)}$ = 2.63, p=0.021; HIPP: average Spearman r = 0.19, $t_{(13)}$ = 5.75, p<0.001). Next, we repeated the PDC analyses focusing on the 12–16 Hz spindle band and found the previously observed increase in neocortical to hippocampal information transfer around the SW-Rs, but again only for long- and not short-duration ripples (*Figure 4C*).

## Discussion

Memory consolidation is thought to rely on an intricate interplay between SOs, sleep spindles and SW-Rs. While the link between each of these sleep signatures and effective consolidation has been established across species (*Antony et al., 2018*; *Ego-Stengel and Wilson, 2010*; *Girardeau et al., 2009*; *Latchoumane et al., 2017*; *Lustenberger et al., 2016*; *Maingret et al., 2016*; *Ngo et al., 2013*), less is known about their dynamic interactions underlying the purported information transfer from hippocampus to neocortex (*Diekelmann and Born, 2010*). According to the 'active systems consolidation' framework (*Born and Wilhelm, 2012*; *Klinzing et al., 2019*), sleep spindles are thought to open fine-tuned windows of opportunity for cross-regional synchronization and plasticity

(*Antony et al., 2019*; *Born and Wilhelm, 2012*; *Helfrich et al., 2018*; *Niethard et al., 2018*; *Seibt et al., 2017*; *Sejnowski and Destexhe, 2000*). Consistent with this notion, recent work has demonstrated increased functional connectivity during sleep spindles across neighboring cortical regions (*Das et al., 2017*; *Laventure et al., 2018*). Moreover, simultaneous EEG-fMRI recordings showed an increase in functional BOLD connectivity between hippocampus and neocortex during sleep spindles (*Andrade et al., 2011*). However, none of these studies have linked spindle-mediated connectivity to hippocampal ripples, the putative electrophysiological marker of memory reactivation (*Buzsáki, 2015*; *Kudrimoti et al., 1999*). In rodents, hippocampal ripples have been shown to coincide with spindles in prefrontal cortex (*Siapas and Wilson, 1998*). Similarly, we and others have shown a temporal nesting of SW-Rs within spindles observed in human hippocampus or parahippocampal areas (*Clemens et al., 2007*; *Helfrich et al., 2019*; *Jiang et al., 2019a*; *Staresina et al., 2015*). However, it has remained unclear whether hippocampal-neocortical communication indeed increases during hippocampal ripples and what the directional dynamics of such interactions might be. Here, we analyzed EEG data acquired during NREM sleep both in hippocampus (HIPP) and at scalp electrode Cz (*Figure 1*), the latter integrating across fronto-parietal neocortical (NC) networks. Time-locking the continuous EEG to individual HIPP SW-Rs, we first showed that spindle power (~12–16 Hz) increases not only in HIPP around SW-Rs (*Jiang et al., 2019a*; *Jiang et al., 2019b*; *Staresina et al., 2015*), but also in NC (*Figure 2*). More importantly, using (directional) coherence analyses (and thus moving beyond mere co-occurrences), we were able to show a ripple-locked increase in (directional) NC-HIPP coupling in the spindle band (*Figure 3*). These results dovetail with a recent iEEG study by *Helfrich et al., 2019*, who showed a ripple-locked, driving influence from electrode contacts in PFC to contacts in the medial temporal lobe (MTL) in the spindle band. While MTL contacts in that study also included entorhinal cortex, perirhinal cortex and parahippocampal cortex, we here pinpoint the influence from NC directly to posterior hippocampus. Together, these results support the idea that spindles serve as the mechanistic vehicle to synchronize the hippocampus with neocortical target sites at the time of hippocampal memory reactivation (*Antony et al., 2019*).

It deserves explicit mention though that we did not employ a pre-sleep learning task, rendering the notion of hippocampal memory reactivation during SW-Rs rather speculative. That said, recent rodent work demonstrated that specifically long-duration ripples are linked to neural replay and are particularly critical for memory consolidation (*Fernández-Ruiz et al., 2019*). If our observed hippocampal-cortical interactions reflect - at least in part - mnemonic information transfer, they should thus be more pronounced during longer relative to shorter ripples. Indeed, as shown in *Figure 4*, we found a selective increase in hippocampal-neocortical communication during long-duration compared to short-duration ripples. Nevertheless, conclusive evidence for memory reactivation would of course require well-designed memory tasks around the sleep recordings.

An important remaining question is how this hippocampal-neocortical dialogue is initiated in the sleeping brain, that is what governs the interplay between spindles and SW-Rs. One intuitive scenario is that upon spontaneous SW-Rs in HIPP, sleep spindles – projected from the thalamus to both HIPP and NC – synchronize the two regions. Indeed, rodent recordings have shown that prefrontal cortex neurons fire after HIPP neurons during slow wave sleep (*Siapas and Wilson, 1998*), and the stronger the hippocampal firing burst, the more likely a spindle event is to be observed in PFC (*Wierzynski et al., 2009*). An alternative scenario, however, is that HIPP-NC interactions are initiated in neocortex, perhaps ensuring that cortical target sites are in a state conducive to plasticity (*Rothschild, 2019*). Rodent recordings from NC and HIPP have shown that the NC spindles indeed trigger HIPP firing and associated ripples (with NC spindles emerging ~200 ms before HIPP ripples *Sirota et al., 2003*) and are prevalent in both regions thereafter (*Peyrache et al., 2009*). Moreover, a recent rodent study showed evidence for a cortical-hippocampal-cortical loop. Engagement of task-relevant NC sites first predicted neural activation patterns of HIPP assemblies by ~200 ms, and activation patterns in HIPP in turn predicted activation in NC (*Rothschild et al., 2017*). Our current results show a remarkable overlap with these studies regarding the onset of NC-HIPP coupling relative to HIPP ripples (~200 ms; *Figure 3D and E*), highlighting the role of spindles in synchronizing NC and HIPP prior to SW-Rs. Moreover, we were further able to show a directional influence of NC to HIPP. In particular, as shown in *Figure 3D*, partial directed coherence (PDC) pointed toward top-down influence from NC to HIPP in the spindle band. Corroborating the PDC analysis, we show that the prevalence of spindle onsets rises significantly in both HIPP and NC prior to ripples, but does so

sooner in NC than in HIPP (*Figure 3E*). In sum, these results suggest that the spindle-mediated hippocampal-neocortical dialogue is initiated in NC and is sustained for an extended period after the SW-Rs, allowing for the hypothesized transfer of memory traces reactivated during HIPP SW-Rs (*Figure 3F*).

Interestingly, in the rodent study mentioned above (*Rothschild et al., 2017*), the authors were also able to bias the NC-HIPP-NC loop experimentally by playing sounds related to wake learning episodes. Specifically, presentation of the sound triggered cortical representations in relevant NC sites, which predicted hippocampal patterns during SW-Rs. The hippocampal signal in turn predicted neocortical activation patterns (although the sound was no longer present). This casts an intriguing light on some recent targeted memory reactivation (TMR) studies in humans (*Cairney et al., 2018*; *Schreiner et al., 2018*). Using multivariate decoding methods on scalp EEG data, these studies indicated two phases of memory-related reinstatement after auditory cues: An initial peak within the first second and another peak at ~2 s after cue onset, with concomitant increases in spindle power around both peaks (*Cairney et al., 2018*). In light of the results reported in *Rothschild et al., 2017* and in conjunction with our current findings, one possibility is that hippocampal reactivation occurs between the two cortical spindle/reactivation events. Indeed, examination of NC spindle onsets around HIPP ripples hints toward a biphasic spindle increase (one before and one after the HIPP ripple; *Figure 3E*), although the second peak did not reach significance compared to the control events. A recent comprehensive examination of cortical spindles and their temporal alignment to hippocampal ripples showed that spindles in some cortical areas precede hippocampal ripples, whereas spindles in other areas follow them (*Jiang et al., 2019a*; *Jiang et al., 2019b*). While the pre-ripple spindles may thus reflect a first (coarse) cortical initiation of a reactivation event, post-ripple spindles in other cortical regions might reflect more detailed reinstatement following the hippocampal contribution. In particular, hippocampal reactivation during SW-Rs might enrich the cortically reactivated memory trace with spatio-temporal episodic details (*Lewis and Bendor, 2019*; *Rothschild, 2019*). This idea is also consistent with a recent TMR study that revealed a cortical spindle refractory period during which the delivery of reminder cues was relatively ineffective (*Antony et al., 2018*). In other words, hippocampal engagement between two cortical spindles might be dedicated to mnemonic (re)processing, resulting in limited resources for the processing of additional memory cues.

While our interpretation of results has been largely guided by the active systems consolidation framework (*Klinzing et al., 2019*), some of our findings are also consistent with alternative accounts. For instance, in line with the synaptic homeostasis hypothesis (*Tononi and Cirelli, 2014*), a recent rodent study showed that hippocampal SW-Rs trigger synaptic depression (*Norimoto et al., 2018*). Moreover, experimental perturbation of ripple activity not only halted synaptic down-regulation but also lead to an impairment of memory consolidation. Accordingly, rather than initiating memory transfer, hippocampal SW-Rs might shape memories, brought online via preceding NC spindles, by pruning irrelevant mnemonic elements. The refined memory trace might then be transferred back to neocortical sites via co-activation in the spindle band.

To conclude, our findings suggest that – ignited by neocortex - sleep spindles mediate cortical-hippocampal communication around SW-R. However, to what extend this interaction might underlie the reorganization of hippocampal information to neocortical structures for long-term memory or a synaptic down-scaling remains an intriguing open question for future research.

## Materials and methods

### Participants

EEG data from 14 patients (35.4 ± 3.0 years of age, seven females) suffering from pharmacoresistant epilepsy were analyzed, which were recorded at the Department of Epileptology, University of Bonn. All patients gave informed consent, the study was conducted according to the Declaration of Helsinki and was approved by the ethics committee of the Medical Faculty of the University of Bonn. Intracranial depth electrodes for presurgical evaluation of seizure onset zones were implanted stereotactically, either via the occipital lobe along the longitudinal axis of the hippocampus or laterally via the temporal lobe. Implantations of depth electrodes were bilateral but only electrodes from the non-pathological hemisphere (according to clinical monitoring) entered the analyses. Intracranial

recordings were obtained continuously for the duration of the patient's stay, but polysomnography was restricted to a single night to mitigate possible discomfort caused application of additional scalp electrodes. We focused our analyses on the posterior hippocampus as functional coupling has been shown between thalamus and posterior, but not anterior hippocampus (*Fanselow and Dong, 2010*; *Poppenk and Moscovitch, 2011*; *Zarei et al., 2013*). Accordingly, increased spindle density and spindle-ripple coupling have been reported for posterior and not anterior hippocampus (*Jiang et al., 2019a*; *Jiang et al., 2019b*; *Staresina et al., 2015*). The selection of the hippocampal contact used for analyses was based on anatomical and functional criteria. First, contacts in posterior hippocampus were marked based on post-surgical MRI scans. If no post-surgical MRIs were available (*n* = 3), we based our designation to posterior hippocampus on the surgical implantation scheme. Among the resulting posterior hippocampal contacts, we took forward the contact with the highest spindle density, that is total number of algorithmically identified spindle events divided by the amount of artifact-free NREM sleep. *Table 3* lists the MNI coordinates for the included contacts.

Note that although the data analyzed here have been used in previously publications, the current study was guided by different questions and all analyses reported here are novel. Specifically, *Staresina et al., 2015* we reported intrahippocampal dynamics between SOs, sleep spindles and ripples, but did not focus on ripple-locked inter-regional dynamics. *Cox et al., 2020* and *Cox et al., 2019* examined the overall coupling between a wide range of frequencies and regions using phase-based approaches on the continuous signal, rather than on ripple-locked events. In short, none of these previous studies examined functional interactions between hippocampus and neocortex time-locked to SW-Rs, nor the specific role of sleep spindles in mediating these interactions.

## EEG recordings and pre-processing

Depth EEG recordings were referenced to linked mastoids and acquired with a sampling rate of 1 kHz (bandpass filter: 0.01 Hz (6 dB per octave) to 300 Hz (12 dB per octave)). For the sleep recordings, additional electrodes were placed on participants' scalps at positions Cz, C3, C4 and Oz according to the 10–20 system. Electro-ocular activity (EOG) was recorded at the outer canthi of both eyes and submental electromyographic activity (EMG) was acquired with electrodes attached to the chin. Electrode impedances were all below 5 kΩ.

Sleep stages were determined visually using scalp EEG, EOG, and EMG recordings for consecutive 20 s epochs according to standard criteria (*Rechtschaffen and Kales, 1968*). For each night, the

**Table 3.** MNI coordinates of the included electrode contacts.
For three patients, no MRI was available.

| Patient | MNI coordinates | | |
| --- | --- | --- | --- |
| | X | Y | Z |
| 1 | −26 | −29 | -8 |
| 2 | | | |
| 3 | | | |
| 4 | 26 | −28 | -7 |
| 5 | | | |
| 6 | 27 | −29 | -3 |
| 7 | −25 | −28 | 0 |
| 8 | −33 | −33 | -2 |
| 9 | 27 | −31 | -9 |
| 10 | 28 | −35 | -7 |
| 11 | −29 | −38 | -9 |
| 12 | 29 | −31 | 3 |
| 13 | 32 | −30 | -1 |
| 14 | −28 | −34 | -5 |

proportion of sleep stages S1, S2, SWS (i.e. S3 and S4) and REM sleep were calculated relative to the total time spent asleep (*Table 1*).

For pre-processing, an automated algorithm was applied to identify three different types of artifacts separately for each sleep stage: First, on a 0.3 to 150 Hz band-pass filtered signal, amplitude-based artifacts were scored as values exceeding ±750 μV based on previous work (*Axmacher et al., 2008*). Next, we identified gradient artifacts, that is strong deflections in the signal caused in particular by interictal spikes. Based on the 0.3 to 150 Hz band-pass filtered signal we first calculated for each time-point the difference in amplitude to the next time-point. This difference signal was then used to derive an individual threshold determined by its median ±6 * interquartile range across all time points. Accordingly, whenever the difference signal exceeds this threshold a gradient artifact was scored. To identity high-frequency bursts emerging from arousals or movement, the EEG signal was high-pass filtered at 150 Hz and the root mean square (RMS) signal calculated based on a window length of 100 ms. Again, an individual threshold was determined by the median + 4 * interquartile range of the RMS signal and time points were marked as high frequency burst if the RMS signal exceeded the corresponding threshold for at least 100 ms. All detected artifact samples were then padded by ±250 ms. Furthermore, artifact-free intervals shorter than 3 s were also marked as artifacts. The threshold used are based on previous publications (*Axmacher et al., 2008*; *Staresina et al., 2015*) and automated detection was followed by a visual inspection.

## Offline detection of discrete spindle and ripple events

Discrete spindles and ripples were detected during artifact-free NREM sleep using offline algorithms (*Staresina et al., 2015*). For spindle detection, the NC- and HIPP-signals were band-pass filtered at 12–16 Hz and the root mean square signal (RMS) was calculated based on a 200 ms windows followed by an additional smoothing with the same window length. A spindle event was identified whenever the smoothed RMS-signal exceed a threshold, defined by the mean plus 1.25 times the standard deviation of the RMS-signal across all NREM data points, for at least 0.4 s but not longer than 3 s. Importantly, time points exceeding an upper threshold determined by the mean RMS-signal plus 5 times its the standard deviation were excluded (*Stark et al., 2014*). The upward and downward threshold crossings represent the onset and end of a spindle event. Sleep spindles are commonly separated into frontally dominant slow (9–12 Hz) spindles and centro-parietal fast spindles (12–15 Hz) (*De Gennaro and Ferrara, 2003*). Here, we focus on fast spindles, which have been consistently shown to nest SW-Rs (*Jiang et al., 2019a*; *Jiang et al., 2019b*; *Staresina et al., 2015*) and, specifically in conjunction with SOs, contribute to plastic changes (*Niethard et al., 2018*) and sleep-dependent memory consolidation (*Cairney et al., 2018*; *Ngo et al., 2013*).

Detection of discrete ripple events in the hippocampal depth recordings followed the same procedure, except that the EEG signal was band-pass filtered from 80 to 120 Hz, encompassing the ripple frequency at ~90 Hz in humans (*Bragin et al., 1999*; *Helfrich et al., 2019*; *Jiang et al., 2019a*; *Staba et al., 2002*; *Staresina et al., 2015*), and both RMS calculation and smoothing were based on 20 ms windows. Detection and upper cut-off threshold were defined by the mean of the RMS-signal plus 2.5 or 9 times the standard deviation, respectively. Potential ripple events with a duration shorter than 38 ms (corresponding to 3 cycles at 80 Hz) or longer than 500 ms were rejected. Additionally, all ripple events were required to exhibit a minimum of three cycles in the raw EEG signal. For subsequent analysis, all ripple events were segmented into 2 s epochs centered on their positive peak.

Finally, to ensure that both spindle and ripple events were not caused by spurious broadband power increases but reflect discrete events within our frequency range of interest, we implemented a routine to discard false positives based on their frequency profile. To this end, we calculated a time-frequency representation time-locked to the maximum of each spindle or ripple event (spindles: frequencies from 9 to 19 Hz in 0.5 Hz steps, time window of ±750 ms in 2 ms steps; ripples: frequencies from 65 to 135 Hz in 2 Hz steps, time window of ±100 ms in 2 ms steps) and extracted the frequency profile by averaging along the time dimension from −0.5 to +0.5 s for spindle events or from −0.05 to +0.05 for ripple events. A spindle/ripple event was rejected as a false positive whenever the frequency profile did not exhibit a prominent peak, that is a decline in amplitude on both sides of at least 20% with respect to its maximum value (determined with the prominence output of the MATLAB function 'findpeaks'), within the frequency range of interest, that is, between 12 and 16 Hz or 80 and 120 Hz, respectively.

## Ripple-free control events

To statistically assess of the dynamics of spectral power and functional connectivity around sharp-wave ripples, we used carefully matched control events in which no ripples were observed but without any restrictions on the occurrence of sleep spindles (see e.g. *Cox et al., 2020*; *Helfrich et al., 2019*; *Klinzing et al., 2016* for a similar approach). One advantage of this procedure is that it avoids the arbitrary decision of what constitutes a proper pre-ripple baseline period. In particular, for each participant's *n* observed ripple events, we derived *n* non-ripple events, that is artifact-free NREM epochs matching the duration of each individual event including an additional padding of 1.5 s before and after in which our ripple detection algorithm did not indicate the presence of a ripple. Furthermore, to ensure that signal properties are maximally matched between target events and surrogates, control events were only drawn from a 10 min time window before and after the corresponding ripple event. The probability underlying the randomized selection of control events within such a 10 min interval was modulated according to a normal distribution. Epochs once assigned to control events were discarded from subsequent iterations to exclude overlapping non-events. This procedure was repeated 100 times. These 100 sets of matched control events were averaged for power comparisons (*Figure 2* and *Figure 2—figure supplement 1*), that is each empirical ripple event is paired with one control event averaged across the 100 iterations, allowing for paired t-tests. For connectivity metrics, which are derived across events, the 100 iterations were used to build a control event distribution of 100 values against which to compare the empirical value (*Figure 3*, *Figure 3—figure supplements 1–3*). Furthermore, it is important to note that the choice of detection criteria influences the absolute number of detected events (see also *Staresina et al., 2015*), which in turn impacts any subsequent analyses. Using matched control events thus mitigates the potential bias by differences in trial numbers and represent a stringent statistical assessment for functional connectivity measures. Visualization of spindle and sharp-wave ripples waveform as shown in *Figure 1* was performed by averaging the EEG signals time-locked to the minimum spindle trough or maximum ripple peak segmented into ±1 s and ±0.5 s epochs, respectively.

## Time-frequency representations

Ripple-locked TFRs were calculated on the epoched ripple-events across all patients using Morlet wavelets for frequencies from 1 to 20 Hz with a 0.5 Hz resolution in 20 ms steps. For frequencies $\geq$ 5 Hz, the number of cycles was set adaptively to half of the corresponding frequency (or rounded up to the next integer value) but at least 5 cycles, resulting in time windows of approximately 500 ms. For frequencies below 5 Hz, that is 1 to 4.5 Hz, cycle numbers were reduced to values ranging from 2 to 4, reducing the window size and thereby increasing availability of artifact-free segments. TFRs on control events were calculated with an identical procedure. Of note, for the $f_{Spindle}$ = 12–16 Hz spindle range, Gaussian-shaped Morlet wavelets are based on $n_{cycles}$ = 6–8 cycles, which corresponds to a temporal resolution of $\sigma_t$0.167 s on average ($\sigma_t n_{cycles}/(\pi \cdot f_{Spindle})$) (*Tallon-Baudry et al., 1996*). Note that assessment of the temporal order of spindle onset times (*Figure 3E*) is based on spindle events detected with a procedure not involving wavelets (see above).

To correct statistical analysis for multiple comparisons, a cluster-based permutation procedure was applied as implemented in FieldTrip (*Oostenveld et al., 2011*), using a cluster threshold of p<0.05 and a final threshold for significance of p<0.05.

## Coherence

Following the region-specific TFR analyses highlighting overall power changes based on an average across all ripple events, we next assessed the degree of functional connectivity between the neocortex and hippocampus around SW-Rs on a trial-by-trial basis by examining spectral coherence. Time-resolved coherence from −1 to +1 centered on ripple events was calculated for frequencies from 1 to 20 Hz using cross-spectral densities from complex time-frequency representations (ft_connectivityanalysis function with method-parameter = 'coh' in FieldTrip). Statistical analysis was initially restricted to the concurrent spindle increases in HIPP and NC obtained from the previous TFR analyses, which encompasses a time interval from −220 to 460 ms around the ripple maximum and a frequency range between 11.5 to 18 Hz. To this end, the empirical value for coherence around ripples was obtained by averaging across our window of interest and tested for significance by means of a

z-statistics using the mean and standard deviation of the corresponding coherence from the 100 control datasets:

$$z = (\mathrm{Coh}_{ripple} - mean(\mathrm{Coh}_{control})) \, / \, std(\mathrm{Coh}_{control})$$

The z-values were than transformed into p-values, with a significance threshold of z-values greater or smaller than +/- 1.96. Furthermore, to illustrate the temporal dynamics of coherence in the spindle range, we first averaged the resulting coherence representation across the frequency dimension from 12 to 16 Hz to obtain a coherence time series and then performed a z-transformation with the corresponding time series from the control data.

## Partial directed coherence

To examine directionality in the cortical-hippocampal communication we calculated partial directed coherence (PDC, ft_connectivityanalysis function with method-parameter = 'pdc' in Fieldtrip). PDC extends the concept of coherence of mutual synchrony, by decomposing the underlying temporal relationship into a directional influence between the regions of interest, that is, to what extent does the past of NC predict present activity in HIPP and vice versa. The concept of PDC reflects a frequency-domain representation of Granger causality (*Baccalá and Sameshima, 2001*). In order to resolve directionality in time, we computed PDC on complex power spectra obtained on 512 ms long intervals shifted from −1 to +1 s around detected ripple events in 20 ms steps. Subsequently, statistical analyses focused on mean PDC values obtained from the significant cluster reflecting the conjoint spindle increases in HIPP and NC. Again, to illustrate the temporal evolution of PDC in the spindle range, we first extracted the corresponding time series by averaging from 12 to 16 Hz and then calculated a z-score with respect to the control data. Finally, we subtracted the resulting z-scores for the two directions, that is NC driving HIPP minus HIPP driving NC. Thus, positive values indicate a directional influence of neocortical on hippocampal spindle activity and vice versa for negative values. Given the ambiguity of any directional influence in the absence of functional coupling, we restrict display and interpretation of PDC to time/frequency bins exhibiting positive spectral coherence in the preceding analysis. Note that PDC calculations employed here follow a non-parametric approach, that is, they are not based on a previous autoregressive (AR) model of the EEG signals but are directly derived from wavelet-based methods. This approach was chosen due to lacking an a priori hypothesis for the AR order, and more importantly, to base all our analyses on the same wavelet framework.

## Amplitude- and phase-based connectivity

Our examination of mutual and directed connectivity based on spectral and partial directed coherence has the advantage that both rely on the same mathematical framework and thus allow for internally consistent statistical assessments (i.e. PDC informing on the directionality of the observed spectral coherence). Nonetheless, we asked to what extend hippocampal-neocortical communication is mediated by cross-regional amplitude or phase-relationships. To disentangle this, we additionally calculated the power-power correlation (*Hipp et al., 2012*) and phase locking value (PLV, *Lachaux et al., 1999*) between HIPP and NC during ripples and control events.

It is important to note that connectivity measures are prone to volume conduction (despite long distances between regions of interest) or spurious coupling effects introduced by a common reference. To mitigate this concern, we removed signal components exhibiting identical instantaneous phase by orthogonalizing our data before calculating the power-power correlation between HIPP and NC (*Hipp et al., 2012*). More specifically, for each ripple/control event, the complex TFR for NC was orthogonalized with respected to HIPP (NC ⊥ HIPP) and HIPP orthogonalized with respect to NC (HIPP ⊥ NC) prior to computing the absolute power. Next, for each time-frequency point we calculated the correlation between NC and (HIPP ⊥ NC) and between (NC ⊥ HIPP) and HIPP across all trials and averaged both resulting correlation maps.

Finally, we examined phase-locking between HIPP and NC (*Lachaux et al., 1999*). Based on complex TFRs for frequencies from 1 to 20 Hz and between −1 s and +1 s around ripple/control events, the PLV (ft_connectivityanalysis function with method-parameter = 'plv' in Fieldtrip) extracts the phase information and evaluates the variation in phase-difference between HIPP and NC for a given frequency and time point across all ripple and control events, respectively.

## Peri-event histograms of spindle onsets

Finally, given our findings on the directional influence between NC- and HIPP-spindles we inspected the timing of spindle onsets within both regions of interest. To this end, we created peri-event histograms (bin size = 50 ms) of the onsets of discrete spindle events time-locked to ripple and control events occurring between −0.5 and +0.5 s. The resulting histograms were normalized by the total number of detected spindle onsets (multiplied by 100). Identical to our previous analyses, statistical comparison with the control data was performed by calculating a z-value with respect to the mean and standard deviation of the control distribution for each region and bin. Note that discrete spindle events may co-occur with more than one SW-R, resulting in multiple inclusion in the peri-event histogram. However, the percentage of such events is ~0.2% and thus negligible.

# Additional information

### Funding

| Funder | Grant reference number | Author |
| --- | --- | --- |
| Wellcome | 107672/Z/15/Z | Bernhard Staresina |
| Deutsche Forschungsgemeinschaft | SFB1089 | Juergen Fell |

The funders had no role in study design, data collection and interpretation, or the decision to submit the work for publication.

### Author contributions

Hong-Viet Ngo, Formal analysis, Writing - original draft; Juergen Fell, Conceptualization, Writing - original draft; Bernhard Staresina, Conceptualization, Writing - review and editing

### Author ORCIDs

Hong-Viet Ngo  https://orcid.org/0000-0001-5828-5588
Bernhard Staresina  https://orcid.org/0000-0002-0558-9745

### Ethics

Human subjects: Informed consent was obtained from all patients and the study was approved by the ethics committee of the Medical Faculty of the University of Bonn (reference number 042/16).

### Decision letter and Author response

Decision letter https://doi.org/10.7554/eLife.57011.sa1
Author response https://doi.org/10.7554/eLife.57011.sa2

# Additional files

### Supplementary files

• Transparent reporting form

### Data availability

Raw EEG data (from NC and HIPP) and all results presented in all figures have been uploaded to the Open Science Framework (https://doi.org/10.17605/OSF.IO/3HPVR). Furthermore, all Matlab code used for data analysis is publicly available on a GitHub repository (https://github.com/episodicmemorylab/Ngo_et_al_eLife2020; copy archived at https://github.com/elifesciences-publications/Ngo_et_al_eLife2020).

The following dataset was generated:

| Author(s) | Year | Dataset title | Dataset URL | Database and Identifier |
|---|---|---|---|---|
| Ngo H-V, Fell J, Staresina B | 2020 | Sleep spindles mediate hippocampal-neocortical coupling during long-duration ripples | https://doi.org/10.17605/OSF.IO/3HPVR | Open Science Framework, 10.17605/OSF.IO/3HPVR |

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
