## [Decision Letter]

**Acceptance summary:**

The study shows that frontal spindles (as measured at scalp) trigger hippocampal activity, which is thought to reflect mnemonic content. The study is timely as it contributes to a growing number of publications which suggest that the hippocampal--neocortical dialogue is not solely driven by the hippocampus, but heavily relies on bidirectional interactions. In addition, the authors report differential connectivity effects for short and long duration ripples in humans, in line with recent rodent data.

**Decision letter after peer review:**

Thank you for submitting your article "Sleep spindles mediate hippocampal-neocortical coupling during (long-duration) sharp-wave ripples" for consideration by *eLife*. Your article has been reviewed by Laura Colgin as the Senior Editor, a Reviewing Editor, and three reviewers. The following individuals involved in review of your submission have agreed to reveal their identity: Markus Werkle-Bergner (Reviewer #2); Sara Mednick (Reviewer #3).

The reviewers have discussed the reviews with one another and the Reviewing Editor has drafted this decision to help you prepare a revised submission.

Specifically, we are asking editors to accept without delay manuscripts, like yours, that they judge can stand as *eLife* papers without additional data, even if they feel that they would make the manuscript stronger. Thus the revisions requested below only address clarity and presentation. Especially, please note (and address adequately throughout the manuscript) the potential concern w.r.t. to potential overlap/incrementality to other papers the research group has published or is about to publish on these data (see Essential revisions #1 below).

Overall, this is a timely manuscript exploring an important physiological mechanisms in humans that was so far only demonstrated in animal models. The statistical and signal processing methods are sound. In addition, the manuscript is very well written with a nicely balanced take on the strength and weaknesses of the experimental approach taken. The first description of different effects of short and long ripples in humans constitutes the main novelty of the paper.

Summary:

In the present manuscript, Ngo et al., report an extensive reanalysis of a previously published dataset (Staresina et al., 2015), which collectively suggests that frontal spindles (as measured at scalp electrode Cz) trigger hippocampal activity, which is thought to reflect mnemonic content. The study is timely and interesting and contributes to a growing number of publications that suggest that the hippocampal-neocortical dialogue is not solely driven by the hippocampus, but heavily relies on bidirectional interactions.

In addition, the authors report differential connectivity effects for short and long duration ripples, in line with a recent paper from the Buzsaki lab that suggested distinct roles for memory formation in rodents.

Essential revisions:

1) Even though the authors present their results as novel, the majority of reported effects has been featured in several recent publications, which should be discussed in the correct context.

In particular, it is difficult to extract the main findings if one considers several other publications from the same dataset: Staresina, 2015, Cox, 2019 (heterogeneous spindle profiles) and Cox, 2019 (phase-based coordination).

The authors state that the intent of current study was 'To examine whether SW-Rs co-occur not only with spindles in HIPP (Staresina et al., 2015) but also in NC', which is also what has been described in the two papers by Cox et al.

2) Authors say they use matched surrogate data, and point out a couple of advantages, which may potentially be real. But there are significant advantages of surrogate data that is developed over 1000s of iterations, as well. Is this really surrogate data, or control data, i.e., periods of signal without the features of interest (non-ripple events)? Can the authors compare their methods with standard surrogate methods and show the same results? Also, is it correct to call this surrogate, or is it control data?

---

## [Author Response]

Essential revisions:1) Even though the authors present their results as novel, the majority of reported effects has been featured in several recent publications, which should be discussed in the correct context.In particular, it is difficult to extract the main findings if one considers several other publications from the same dataset: Staresina, 2015, Cox, 2019 (heterogeneous spindle profiles) and Cox, 2019 (phase-based coordination).The authors state that the intent of current study was 'To examine whether SW-Rs co-occur not only with spindles in HIPP (Staresina et al., 2015) but also in NC', which is also what has been described in the two papers by Cox et al.

We agree that it is important to more clearly delineate our current findings vis a vis related recent publications. In brief, none of the mentioned studies examined ripple-locked hippocampal-neocortical interactions and none distinguished long- vs. short-duration ripples. In Staresina et al., (2015), we reported ripple-locked intrahippocampal dynamics, but not ripple-locked hippocampal-neocortical interactions. In Cox et al., (2019), we explored phase-amplitude coupling in the continuous (i.e., not event-locked) signal across a wide range of frequencies, regions and across sleep stages. No functional interactions between hippocampus and neocortex were examined and no ripple-locked analysis was conducted. In Cox et al., (2019, 2020), we investigated phase-based communication between hippocampus and lateral temporal cortex, and again cross-frequency coupling was examined in the continuous signal and not locked to ripples.

We now explicitly highlight these differences to previous publications based on the same dataset in the Materials and methods section:

Subsection “Participants”:

“[…] Table 3 lists the MNI coordinates for the included contacts. Informed consent was obtained from all patients and the study was approved by the ethics committee of the Medical Faculty of the University of Bonn.”

Note that although the data analyzed here have been used in previously publications, the current study was guided by different questions and all analyses reported here are novel. Specifically, in Staresina et al., (2015) we reported intrahippocampal dynamics between SOs, sleep spindles and ripples, but did not focus on ripple-locked inter-regional dynamics. Cox et al., (2019 and 2020) examined the overall coupling between a wide range of frequencies and regions using phase-based approaches on the continuous signal, rather than on ripple-locked events. In short, none of these previous studies examined functional interactions between hippocampus and neocortex time-locked to SW-Rs, nor the specific role of sleep spindles in mediating these interactions.

2) Authors say they use matched surrogate data, and point out a couple of advantages, which may potentially be real. But there are significant advantages of surrogate data that is developed over 1000s of iterations, as well. Is this really surrogate data, or control data, i.e., periods of signal without the features of interest (non-ripple events)? Can the authors compare their methods with standard surrogate methods and show the same results? Also, is it correct to call this surrogate, or is it control data?

We thank the reviewers for the excellent suggestion to more appropriately use the term ‘control events’, which we have now adopted throughout the manuscript. Note that this is a commonly used approach when examining sleep event-based effects (e.g., Helfrich et al., 2019; Klinzing et al., 2016) and we now make explicit reference to other studies using this approach in the Materials and methods section. In general, we believe that our implementation of this control events approach is particularly well-suited for the current analyses for a number of reasons. First, we designed our algorithm such that for each target event (i.e., ripple), non-overlapping ripple- and artifact-free intervals are detected in close temporal proximity. This ensures that overall signal properties are carefully matched, with the only difference being the presence vs. absence of a hippocampal ripple. It is therefore arguably the most stringent approach to query the role of ripples for hippocampal-cortical dynamics. Second, and specifically for time-frequency based analyses, the control event approach relieves the researcher from the burden of choosing an appropriate pre-event baseline interval, a choice that is usually somewhat arbitrary. Third, basing all analyses on the same set of control events allows us to use a unified statistical framework throughout, i.e., for ripple-locked TFRs (Figure 2) as well as for connectivity analyses (Figure 3).

Nevertheless, we followed the reviewers’ suggestion and repeated the coherence analyses by constructing a more conventional surrogate distribution. Specifically, we held hippocampal intervals constant but shuffled NC intervals prior to connectivity analysis. This means that HIPP and NC data do not stem from the same time period any longer. This shuffling procedure was repeated 100 times. In Author response image 1 we plot time-frequency resolved coherence time-locked to the detected ripple events (left) and averaged across all repetitions of the shuffled data (middle). Furthermore, we plot, analogous to Figure 3A in our original manuscript, the histogram of the coherence for the shuffled data and the observed NC-HIPP coherence based on the central spindle cluster (right). These results indicate that shuffling the NC-HIPP allocation might render the statistical comparisons in the current data too liberal.

**Author response image 1. sa2fig1:** Time-frequency resolved NC-HIPP coherence obtained for observed ripple events (A) and after shuffling the corresponding NC-intervals and averaged across 100 realizations (B). (C) HIPP-NC coherence determined for the cluster of concurrent spindle power around HIPP SW-Rs (see Figure 2). The red line depicts the observed coherence, whereas the histogram depicts the distribution of individual coherence values obtained from the 100 sets of shuffled data.

On a final note, it deserves mention that our ripple-free control events were also based on repeated sampling as the reviewer suggests. This is now described in more detail in the Materials and methods section. In particular, we generated 100 unique sets of control data, which were either averaged for direct TFR comparison with real events (Figure 2) or retained to build a distribution against which the empirical values were compared. 100 (rather than 1000s of) iterations were used because we wanted to keep the set of ripple-free intervals in each iteration non-overlapping with the other sets.